# The Psychological Impact of the Widespread Availability of Palatable Foods Predicts Uncontrolled and Emotional Eating in Adults

**DOI:** 10.3390/foods13010052

**Published:** 2023-12-22

**Authors:** Natália d’Ottaviano Medina, Joana Pereira de Carvalho-Ferreira, Julia Beghini, Diogo Thimoteo da Cunha

**Affiliations:** Laboratório Multidisciplinar em Alimentos e Saúde, Faculdade de Ciências Aplicadas, Universidade Estadual de Campinas, Sao Paulo 13083-872, Brazil; n175095@dac.unicamp.br (N.d.M.); joanacf@unicamp.br (J.P.d.C.-F.); j175563@dac.unicamp.br (J.B.)

**Keywords:** food choice, eating behavior, diet

## Abstract

This study aimed to investigate the role of the psychological impact of environments rich in palatable foods on three aspects of eating behavior: cognitive restraint (CR), uncontrolled eating (UE), and emotional eating (EE). The hypotheses were as follows: (a) The psychological impact (i.e., motivation to eat) of an environment rich in palatable foods will positively predict CR, UE, and EE; (b) dieting will predict CR, UE, and EE; and (c) CR, UE, and EE will positively predict body mass index (BMI). This study had a cross-sectional design in which data were collected online from 413 subjects. The psychological impact of food-rich environments (food available, food present, and food tasted) was assessed using the Power of Food Scale (PFS), and CR, UE, and EE were assessed using the Three-Factor Eating Questionnaire (TFEQ-R18). Both instruments were tested for confirmatory factor analysis. The relationship between constructs was measured using partial least-square structural equation modeling (PLS-SEM). “Food available” positively predicted all TFEQ-R18 factors (*p* < 0.01). “Food present” positively predicted UE (*p* < 0.001) and EE (*p* = 0.01). People currently on a diet showed higher levels of CR (*p* < 0.001) and EE (*p* = 0.02). UE and EE positively predicted BMI. Thus, CR, UE, and EE were positively predicted by the motivation to consume palatable foods in varying proximity, suggesting that the presence of food and, more importantly, its general availability may be important determinants of eating behavior, particularly UE and EE. Health strategies should consider the influence of the food environment to prevent and better manage impairments in eating behavior. Sex differences suggest that special attention should be paid to women. Furthermore, dieting was associated with higher levels of EE, which in turn was associated with higher BMI. Weight loss interventions should consider this vulnerability.

## 1. Introduction

Advances in research on the regulation of food intake, particularly regarding the role of non-homeostatic processes, have provided a new perspective for studying eating behavior [1,2]. As recently suggested, contemporary food intake patterns appear to be strongly determined by sensory, emotional, and cognitive factors [3]. Eating behavior and its determinants play an essential role in this scenario, given the recent increase in eating disorders [4] and obesity [5]. Among the many variables that constitute eating behavior, special attention should be paid to psychological aspects such as cognitive restraint (CR), uncontrolled eating (UE), and emotional eating (EE). There is growing evidence emphasizing harmful associations between high levels of CR, UE, and EE and several health-related factors (e.g., eating habits and attitudes, body image, psychological well-being, and sleep quality [6,7,8,9,10]).

CR is the control exerted over food intake to influence body weight or body shape [11]. It includes behavioral, cognitive, and affective components, such as eating lower-calorie foods or skipping meals, worrying about weight, and negative feelings triggered by the tendency to fail at dieting or maintaining weight loss, respectively [12]. CR appears to be closely related to the internalization of the ideal of thinness and fear of fat, both of which are driven by sociocultural influences [13] and motivate people, especially women, to strive for a thinner body [12]. UE is manifested by general difficulty controlling food intake and the loss of control over food intake, leading to overeating [11]. It has been associated with high energy consumption and increased body weight [14]. EE refers to eating in response to emotional states (such as anxiety or irritability) [15] rather than the typical signs of hunger. Although positive emotions also play an important role, negative emotions seem to have a greater influence on eating behavior [16]. EE generally represents an attempt to reduce emotional discomfort [11], and negative emotions can be recurrent triggers that lead to maladaptive behaviors and overeating [17].

It has been hypothesized that motivation to eat, regardless of physiological needs, is enhanced in environments where highly palatable foods are ubiquitous [18,19]. The widespread availability and easy access to highly palatable and energy-dense foods are evident in today’s food environments [3]. This ‘new environment’ is inextricably linked to colonialism and globalization and reflects the nutritional transition from whole foods to ultra-processed foods that occurred worldwide in the twentieth century [20]. Despite the evolutionary advantage and genetic contribution to seeking out energy-dense foods, such an environment is most likely an issue today as changes in diet have been exacerbated over the past seven decades due to global trade pacts in agricultural commodities, foreign direct investment in food processing, and shifts to niche markets. In Brazil, for instance, sensory appeal is one of the key motivators for food choices [21], and easy access to food via food apps is increasing significantly [22]. In such a context of abundance, the perception of palatable foods is likely to trigger thoughts and cravings for these foods at any time [18].

Given the impact of sensory, emotional, and cognitive factors on contemporary food intake, exploring the role of the food environment on eating behavior is particularly important in light of the increasing number of unhealthy eating patterns that cannot be remedied by homeostatic strategies. Because high levels of CR, UE, and EE are associated with negative health outcomes, a deeper understanding of their determinants would allow for the development of more effective prevention and treatment interventions. The potential psychological impact of environments rich in palatable foods would require strategies that not only target the individual level but also take into account the physical context in which eating behaviors are experienced in an interdisciplinary approach. In the present study, we aimed to investigate the role of the psychological impact (i.e., motivation to eat) of environments rich in palatable foods on CR, UE, and EE. Previous research seems to have only combined these variables when validating psychometric instruments, limiting the ability to examine such relationships in depth. Also, most cross-sectional studies only analyzed simple correlations, which does not allow for the observation of the combined effect of these variables. In an effort to shed light on the determinants of currently relevant aspects of eating behavior, after determining whether and how CR, UE, and EE were predicted by the psychological impact of the modern food environment, we also aimed to understand the role of dieting and BMI in this process.

## 2. Theoretical Background and Model

This study is grounded on current evidence that individuals who are highly influenced by the modern food environment show increased responsiveness to food cues, which may be reflected in part by increased attention to food reward. Previous studies have reported correlations with measures of problematic eating behaviors (e.g., disinhibition, emotional eating, etc.) and possible attempts to moderate such responsiveness to food cues [23]. Thus, we defined three main hypotheses based on a theoretical rationale to achieve the proposed objectives.

(i)The psychological impact (i.e., motivation to eat) of an environment rich in palatable foods will positively predict cognitive restraint, uncontrolled eating, and emotional eating.

Eating regulation depends on both homeostatic and non-homeostatic mechanisms, which are intimately integrated and rely upon common brain structures [2,24,25,26]. CR, UE, and EE seem to be related to non-homeostatic processes, which control food intake in response to emotional, cognitive, and environmental factors and are usually driven by pleasure or sensory perception [1,2].

Dopaminergic reward circuits in the central nervous system are activated not only in response to the consumption of palatable foods [27,28] but also through conditioning, i.e., in response to cues that predict the availability or consumption of such foods [29,30]. It is also known that the high availability of palatable foods and frequent exposure to their cues can influence individuals’ psychological processes, such as thoughts, feelings, and motivation, which in turn can affect food intake [31]. This helps explain why the numerous food cues (e.g., sights, sounds, and smells associated with food) that constitute today’s food environment can motivate consumption and influence what and how much one eats [32]. In such a context of abundance, the perception of palatable foods is likely to trigger thoughts and cravings for these foods at any time [18]. Recently, cravings have been shown to be associated with the widespread availability of cheap, energy-rich, and highly palatable foods [33]. Similarly, motivation to eat palatable foods is expected to predict CR, UE, and EE in an urban sample.

(ii)Dieting will predict cognitive restraint, uncontrolled eating, and emotional eating.(iii)Cognitive restraint, uncontrolled eating, and emotional eating will positively predict body mass index.

Restricting food intake for weight loss (i.e., dieting) is a behavior that is widely studied for its possible paradoxical effects on body weight control. Dieting is a widely used strategy to reduce or control body weight or body shape. There is evidence that cognitive and behavioral constraints lead to weight loss in the short term but may also predict weight gain in the long term [34,35]. Conversely, there is evidence that diets reflect personal vulnerability to weight gain in an urban, obesogenic environment [36].

Although the effects of weight loss over time in people with obesity are not fully understood, metabolic responses may occur, such as a reduction in resting metabolic rate and total energy expenditure [37]. However, particular attention has been paid to studying the psychological and behavioral effects of dieting, which have been linked to binge eating, preoccupation with food, and emotional reactivity [38]. For example, unhealthy behaviors such as skipping meals, using food substitutes, and diet pills in an attempt to lose weight have been shown to predict higher BMI over time in adolescents [39]. Furthermore, longitudinal studies have shown that dieting predicts the occurrence of binge eating [40], especially when dieting is accompanied by depressive symptoms and impaired self-esteem [41]. These findings support the idea that dieting increases the risk of losing control over food. Restrained eating correlates with externalized and disinhibited eating, cravings, and BMI [42]. In addition, EE was higher in former or current dieters than non-dieters [43]. It is important to mention that it is speculated that there is a relevant distinction between rigid or flexible restrictions, with the former considered more linked to disinhibited eating behavior [44].

Dieting is considered a risk factor for EE, especially under stressful conditions [45]. Through self-imposed food restriction, dieters often resist their feelings of hunger, which may interfere with their accurate perception of satiety and increase the risk of eating in the presence of negative emotions [45]. In addition, EE has been consistently associated with the overconsumption of palatable, high-energy foods [16]. Restrained eating and EE appear to be related to psychological aspects (i.e., body satisfaction and self-esteem) and higher BMI and adiposity [46,47]. Based on this evidence, we believe that dieting will predict CR, EE, and UE. CR, EE, and UE will, in turn, predict BMI.

Based on previous findings, a comparison between sexes is also proposed. Growing evidence shows that women have different health beliefs and food avoidance [48]. Women also seek more health-related information, pay more attention to the effects of the goods they buy [49], and place a greater importance on healthy eating than men [50]. CR appears to be closely related to the internalization of the ideal of thinness and fear of fat, both of which are driven by sociocultural influences [13] and motivate people, especially women, to strive for a thinner body [12]. In previous research, women showed higher scores on all factors related to the psychological impact of an environment rich in palatable foods [33,51]. It is, therefore, expected that women would show higher scores for CR, EE, EU, and factors related to the food environment.

The developed hypotheses were tested using partial least-square structural equation modeling (PLS-SEM). PLS-SEM minimizes sample size limitations, makes no distributional assumptions, has higher statistical power than covariance-based methods, and is an appropriate strategy for testing theory [52]. A confirmatory factor analysis (CFA) was first performed to check the measurement quality of all latent constructs used in PLS-SEM.

## 3. Methods

### 3.1. Sample

Participants were recruited online via social media (e.g., Instagram and Facebook) and in person on the university campus. An advertising banner with a QR code and direct message was used for recruitment. The authors of the study carried out the entire recruitment. Men and women were invited to participate in an online cross-sectional study about eating behaviors and palatable foods. To be eligible, participants had to be between 18 and 60 years old and live in an urban area. The sample size was calculated considering the requirements of factor analysis. Since samples with more than 200 participants are suitable for confirmatory factor analysis (CFA), and samples with 300 participants are recommended for multivariate analysis [53], we aimed for a minimum number of 300 people.

Data were collected online via SurveyMonkey for one month (3 November–2 December 2022). A total of 545 Brazilian participants answered the Power of Food Scale (PFS) and the Three-Factor Eating Questionnaire (TFEQ-R18), in addition to sample characterization questions that captured socioeconomic and anthropometric status. Participants with incomplete or monotonous responses (standard deviation = 0 for any scale) were excluded, as were those with diseases that could affect eating habits, such as diabetes, cardiovascular, liver, and kidney diseases. There was no monetary incentive. Thus, the final sample included 413 participants, with a sex balance of 205 women and 208 men. This study was conducted after approval by the University of Campinas Ethics Committee on 5 October 2022 (protocol number: 62848922.7.0000.5404), and all participants gave their informed consent.

### 3.2. Measures 

The PFS is a self-assessment instrument that measures the psychological impact of living in food-rich environments [31]. In other words, the PFS assesses motivation to consume palatable foods [54], especially in environments where they are highly and constantly available [55]. The PFS considers that appetitive reactions can be influenced by the proximity between individuals and palatable foods [31,56]. The questionnaire developed by Lowe et al. [31] consists of 15 items considering changes in appetite responses within three levels of food proximity: food available (widely and easily accessible but not physically present), food present (physically present in the environment but not yet tasted), and food tasted (tasted but not yet consumed) [31]. It is assumed that the higher the total score, the stronger the response to the food environment [57]. All items were measured on a 5-point scale ranging from ‘1—do not agree at all’ to ‘5—strongly agree’. The PFS has been translated and validated for many countries [54,55,58,59]. Its Portuguese version [60] was recently applied to a Brazilian sample with some linguistic adaptations to Brazilian Portuguese and showed adequate factorial structure [51].

The TFEQ proposed by Stunkard and Messick [61] originally consisted of 51 items. It was developed to assess three dimensions of human eating behavior: the cognitive restraint of eating, the disinhibition of control, and susceptibility to hunger. This instrument can verify individuals from extreme voluntary restraint of eating to individuals characterized by a complete lack of eating restraint. Karlsson et al. [11] later proposed a short version of the questionnaire comprising 18 items (TFEQ-R18). In this version, the three factors of eating behavior were divided into CR, UE, and EE. The CR factor refers to controlling food intake behaviors to reduce energy intake and influence body weight and shape. The UE factor combines features of the earlier disinhibition and hunger factors. It reflects the tendency to overeat. The EE factor refers to eating in response to negative emotional states. The response scale consists of a dichotomous, 4-point Likert scale [11]. For the last item of the TFEQ-R18, an 8-point scale was used for restraint (1—no restraint in eating to 8—total restraint). Subjects’ scores for each factor can be classified as absent, low, moderate, or exacerbated using quartiles as cut-off points [62]. In addition, the TFEQ-R18 showed adequate psychometric properties when applied to Brazilian adults [63].

Socioeconomic and anthropometric data collected included age, sex, gender, place of residence, education level, monthly family income, frequency of dieting (5-point scale ranging from ‘1—never’ to 5—always’), currently dieting (yes or no), weight (kg), and height (m). Self-reported measures of weight and height were used to estimate body mass index (BMI). 

### 3.3. Data Analysis

The theoretical distributions of the variables were analyzed using means, variances, skewness, kurtosis, and distribution histograms; the normality of the data was checked using the Kolmogorov–Smirnov test with Lilliefors correction. Although the PFS and TFEQ-R18 are validated instruments already in use in Brazil, they were subjected to a CFA to ensure the quality of latent variables. In addition, CFA is a fundamental first step for structural equation modeling. The CFA was performed according to the original structure of the PFS [31,57] and the 18-item version of the TFEQ [11] using diagonally weighted least squares. Model fit was based on the chi-squared value (*p* < 0.05), the root mean square error of approximation (RMSEA < 0.08), the comparative fit index (CFI > 0.90), the standardized root mean square residual (SRMR < 0.08), the Tucker–Lewis index (TLI > 0.90), and the goodness-of-fit index (GFI > 0.90) [64]. 

The relationship between the constructs was measured using PLS-SEM. We tested a 1st-order model to examine the individual effects of the PFS factors on the TFEQ factors, and it showed significant results consistent with the initially established hypotheses. BMI (continuous) and dieting (binary) were included as observed variables. All indicators validated in the previous CFA were included as latent variables, i.e., food available, food present, food tasted, CR, UE, and EE. The outer model (the part of the model describing the relationships between the latent variables and their indicators) was analyzed using factor loadings (>0.40), composite reliability (>0.70), and the average variance extracted (AVE) (>0.40). The inner model (the part of the model describing the relationships between the latent variables) was analyzed based on the explained variance of the endogenous constructs, with effect sizes classified as small (f^2^ ≥ 0.02), medium (f^2^ ≥ 0.15), or large (f^2^ ≥ 0.35) [65], and predictive relevance (Stone–Geisser’s Q^2^ > 0.15). Multicollinearity was measured by variance inflation factor (VIF) values (<3.3) [66]. Discriminant validity (<0.85) was assessed using the heterotrait–monotrait ratio (HTMT) of correlations [66] A multigroup analysis was conducted to test if men and women had significant differences in their group-specific path coefficients. The bootstrapping procedure with 5000 samples was used to estimate the t-statistics (significance: t > 1.96) and *p*-values (significance: *p* < 0.05) of the estimated loadings.

Differences between the two groups (e.g., women × men) in the PFS and TFEQ mean values were assessed with Student’s *t*-test. Cohen’s d was used to measure the effect size of significant differences, classified as small (d = 0.20), medium (d = 0.50), or large (d = 0.80) [65]. 

A repeated-measure analysis of variance with Bonferroni’s post hoc test was employed to compare multiple paired variables, e.g., to compare each PFS factor.

Statistical analyses were performed using Statistical Package for the Social Sciences v.20 (IBM Corp. Armonk, NY, USA), JASP 0.16.1 (University of Amsterdam, Amsterdam, The Netherlands), and SmartPLS v3.2.8 (SmartPLS GmbH. Bönningstedt, Oststeinbek, Germany).

## 4. Results

### 4.1. Sample

The sample included 413 men and women (50.4 and 49.6%, respectively) who were predominantly white (80.6%) and between 18 and 30 years old (75.5%), with a family income between BRL 3600.01 (USD 676.7) and BRL 7200.00 (USD 1353.4) per month, as shown in Table 1. Although all participants were Brazilian and lived in four of the main regions of the country, 95.4% were from the Southeast, particularly the state of São Paulo (*n* = 375; 90.8% of the total sample), followed by Rio de Janeiro (*n* = 10; 2.4%). The sample’s education level was high, with 69.1% completed or engaged in higher education and 21.3% in postgraduation programs. 

The mean body mass index (BMI) ± standard deviation was 24.8 ± 4.86 kg/m^2^ (range = 16.1 to 54.1 kg/m^2^), with most of the sample classified as normal weight (58.3%) and overweight (26.9%). Although some of the participants reported having rarely (30.3%) or never (27.1%) engaged in dieting, 5.1% of them said, ‘I am always dieting’. When the study was conducted, 23% of the participants were currently dieting, comprising 25% of female and 21% of male participants.

### 4.2. Confirmatory Factor Analysis and Correlations

The CFA for the PFS was performed following the structure presented by Lowe et al. [31] and Cappelleri et al. [57]. The PFS indicators formed three well-defined domains with adequate composite reliability and AVE, as shown in Table 2. The PFS showed adequate fit: χ^2^ = 10,844.5 (*p* < 0.001), RMSEA = 0.06, CFI = 0.98, TLI = 0.98, and GFI = 0.98. All indicators presented adequate factor loadings (Table 2). Factor 2 (food present) and Factor 3 (food tasted) had similar means (3.06 ± 0.98 and 3.06 ± 0.84, respectively), and they were higher than the mean value of Factor 2 (food available) (2.26 ± 091) (P_bonf_ < 0.001 for both). The overall PFS score (aggregate factor) was 2.69 ± 1.34. 

The CFA for the TFEQ-R18 was performed following the structure presented by Karlsson et al. [11] and Martins et al. [63]. The TFEQ indicators formed three well-defined domains with adequate composite reliability and AVE, as shown in Table 3. The TFEQ presented adequate fit: χ^2^ = 11,284.4 (*p* < 0.001), RMSEA = 0.06, CFI = 0.97, TLI = 0.97, and GFI = 0.98. All indicators presented adequate factor loadings (Table 3). Considering the mean values (CR = 2.40 ± 0.77, UE = 2.14 ± 0.57, and EE = 2.25 ± 0.87), CR, UE, and EE were classified as low within the total sample.

All constructs from the TFEQ and PFS had adequate discriminant validity and showed HTMT correlations of < 0.85 between them. Moderate correlations (r > 0.50) were observed among PFS factors (Table 4). UE and EE showed low-to-moderate correlations with PFS factors. On the other hand, CR showed almost null-to-null correlations.

### 4.3. Structural Model

The first-order inner model of PLS-SEM was calculated (Figure 1), and it showed adequate predictive relevance (Q^2^ > 0). The ‘food available’ domain had a positive relationship with all three domains of eating behavior, showing a small effect size predicting CR (f^2^ = 0.01) and medium effect sizes predicting UE (f^2^ = 0.32) and EE (f^2^ = 0.23). The ‘food present’ domain had positive effects, with small effect sizes on UE (f^2^ = 0.08) and EE (f^2^ = 0.01), and it showed no significant effect on CR. None of the paths from the ‘food tasted’ domain were significant. Regarding the aspects of eating behavior, UE and EE domains positively predicted BMI, both with small effect sizes (f^2^ = 0.02; f^2^ = 0.01). CR had no significant effect on BMI or UE. However, CR was predicted by being on a diet, with a small effect size (f^2^ = 0.08). The current diet also showed a small effect size predicting EE (f^2^ = 0.01) but had no significant effect on UE. Furthermore, the CR domain was included as a moderator of the influence of the ‘food present’ domain on UE, and it showed a negative effect (f^2^ = 0.01), diminishing the association between motivation to consume food that is present in the environment and UE.

### 4.4. Differences between Sexes

Several differences were observed between the sexes, as shown in Table 5. Women had higher CR, EE, food available, food present, food tasted, and power of food aggregated factor scores. These differences showed a small effect size (d > 0.2), except for EE, which presented a medium effect size (d > 0.5). There was no significant difference in UE between the sexes. Furthermore, 25% of female and 21% of male subjects were currently on a diet, which does not represent a statistically significant difference (*p* = 0.383).

Despite the differences in the means of several variables, the path coefficients of the PLS-SEM model were similar between the sexes in the multigroup analysis (Table 6). This means that the factors of the PFS predicted eating behavior between sexes similarly.

## 5. Discussion

### 5.1. General Discussion

The main objective of this study was to examine the association between aspects of eating behaviors of significant importance to population health (CR, UE, and EE) and motivation to consume palatable foods in environments where they are highly available (as measured with the PFS). The study also examined the association between these aspects and other variables, such as sex, BMI, and current dieting. Our results suggest that the overall availability of food is a strong and relevant predictor for eating behavior, particularly uncontrolled and EE. Although weaker, the immediate presence of food in the environment (i.e., food present) was also associated with eating behavior. Another important finding was that dieting (which, on the one hand, might be an attempt to protect oneself from today’s wide availability of palatable foods) predicted EE, which was also positively associated with higher BMI in our sample. These results support our original hypothesis that eating behavior is directly influenced by the food environment. Although significant heritability has been demonstrated for appetite traits such as food responsiveness, it has been suggested that the control of environmental conditions may alter the expression of appetite [67]. Therefore, the findings corroborate the key role of managing exposure to highly palatable foods in preventing the learned behaviors associated with weight gain and eating disorders, such as EE [68].

Initially, a CFA was performed for the TFEQ-R18 and PFS. Both questionnaires have shown good fit and reliability indicators when used in several countries [11,31,55,57,60,69,70,71]. The PFS and TFEQ-R18 showed adequate factorial structure with high reliability and internal consistency, consistent with other studies involving the Brazilian population [33,51,63]. Both scales had high factor loadings (>0.40), high CR (>0.70), and adequate AVE (>0.40). Taken together, these CFA results support the finding that the TFEQ-R18 and PFS are stable scales in Brazilian samples.

Among the aspects of eating behavior, CR had the highest mean score, followed by EE and UE, with all three aspects scoring low in the overall sample. However, when comparing between groups, women scored higher on CR and EE than men. These differences support previous findings showing higher levels of CR [72], EE [57], or both [73,74] in female participants. The higher level of CR in females may be primarily due to the social pressure regarding body weight that females face in almost all cultures [75]. These pressures may lead to greater internalization of the thin body, perceived pressure from the media—an orientation toward appearance—and concern about being overweight compared with men [76]. While men in our sample had low scores on CR and EE, scores for women were moderate, and the difference in EE showed a medium effect size. Concerning the PFS factors, food tasted and food present scores were similar and higher than the food available factor score. This result differs from samples from the USA [57] and Iran [58] but is similar to other studies in Brazil [33,51] and Portugal [60]. This highlights the notion that the ‘food present’ factor may be more relevant for Brazilian and Portuguese populations, as these countries share some common cultural factors. We propose to investigate these aspects further. Women also scored higher on all PFS factors, which is consistent with other studies [33,58,77] and supports the idea that women are more susceptible to the hedonic effects of palatable foods.

With adequate predictive relevance, our first-order inner model showed that appetite responses reflecting motivation to eat palatable foods within different levels of food proximity seem to be related to the concepts of CR, UE, and EE. Although, to our knowledge, some associations between PFS factors and aspects of eating behavior have already been reported from the TFEQ [58,59], this was the first time that more detailed analyses, including a predictive model, have been conducted. CR was predicted by the ‘food available’ factor, which suggests that some level of restraint is exerted in response to the mere possibility of accessing palatable foods. This seems to represent a form of general anticipation of today’s food environment. This finding should be carefully scrutinized, as it is well known that while some degree of CR may serve as a protective factor, it is also associated with poorer diets, overeating, and susceptibility to obesity and eating disorders [78]. 

Uncontrolled eating and EE were predicted by the ‘food present’ and ‘food available’ factors. In our model, the path coefficient from ‘food available’ to UE and EE had the largest effect sizes, i.e., not only can the motivational response to eating palatable and physically available food lead to UE and EE, but this response appears to be even greater in a broader context of food availability. This is consistent with hypotheses considering the (so-called obesogenic) environment as the primary driver of eating-related conditions, as it allows exteroceptive stimuli to constantly access the motivational mechanisms underlying the natural drive to eat and can also induce a hyperphagic bias [3]. Thus, this result suggests that avoiding physical proximity (e.g., not storing palatable foods at home) may not prevent disturbances in eating behavior, given the easy and quick accessibility of palatable foods in the modern food environment. Consistent with the recommendation that health interventions should aim to reduce UE and EE rather than increase CR [78], we suggest that not only should individual aspects, such as improving one’s relationship with food, be considered, but also the effects of widespread food availability. Furthermore, CR showed a moderator effect that attenuated the association between motivation to eat palatable physically available foods and UE, whereas it was not directly associated with UE. 

As expected, being on a diet was associated with higher levels of CR. Dieting also predicted higher levels of EE. Because EE was also associated with higher BMI, dieting could be understood as a mediator. Our findings add to the growing body of evidence on the paradoxical effects of dieting. This could be partly explained by previously demonstrated associations between EE and increased amount of food consumed, increased consumption of sweets and desserts, increased food delivery purchase, and decreased consumption of vegetables [6], for example. UE also predicted higher BMI, which appears to be associated with increased amounts of food consumed [6]. Considering previous findings showing that participants who scored lower on all TFEQ domains had the lowest BMI scores, while those who scored higher on UE and EE had the highest BMI scores, it has been suggested that the severity of these aspects of eating behavior may represent an increased risk for overweight and obesity [69]. Thus, although UE may lead to higher BMI, EE, in particular, may be exacerbated by interventions focused on weight loss diets, which may ultimately increase vulnerability to weight gain. Furthermore, all the constructs discussed here were predicted similarly regardless of the sex of the participants. Despite the tendency for women to have higher levels of aspects of eating behavior and greater susceptibility to palatable foods, our model appears to capture how women and men are exposed to the combined functioning between the power of food, CR, UE, EE, dieting, and BMI. 

### 5.2. Limitations of the Study

Our results should be considered in light of several limitations. First, although predictive modeling was used, this study used a cross-sectional design. Longitudinal or experimental studies would allow researchers to confirm these results and draw conclusions about the causal mechanisms between the studied variables. Another limitation is that we did not examine whether participants were taking orexigenic or anorexigenic medications. Finally, the study was web-based, which limited the participation of people with low incomes or from vulnerable areas. Future studies with different sample profiles and designs would contribute to a broader view of the issues discussed here.

## 6. Conclusions

The present study revealed that CR, UE, and EE were predicted by ‘food available’ (widely and easily accessible but not physically present food). The ‘food present’ factor (food physically present) predicted uncontrolled eating and EE. Individuals currently dieting showed higher levels of CR and EE. Finally, UE and EE were associated with BMI. Thus, these aspects of eating behavior were positively related to motivation to consume palatable foods in varying proximity. Our results suggest that the presence of food and, more importantly, its general availability may be an important driver of eating behavior, particularly UE and EE. The observed sex differences also suggest that women are more prone to CR, EE, and the hedonic effects of palatable foods. 

Our research underscores the importance of considering the influence of the contemporary availability of palatable foods in health strategies to prevent and better manage eating behavior impairments with particular attention to women. Although completely changing today’s food environment would be unrealistic, raising awareness of its potentially dangerous role in health is in the public interest. In addition, our findings suggest that dieting (which is typically associated with the intention to control or lose weight) is associated with higher levels of EE, which in turn is associated with higher BMI. Therefore, professionals and health intervention programs should be aware of the potential negative effects of diet-based interventions. 

## Figures and Tables

**Figure 1 foods-13-00052-f001:**
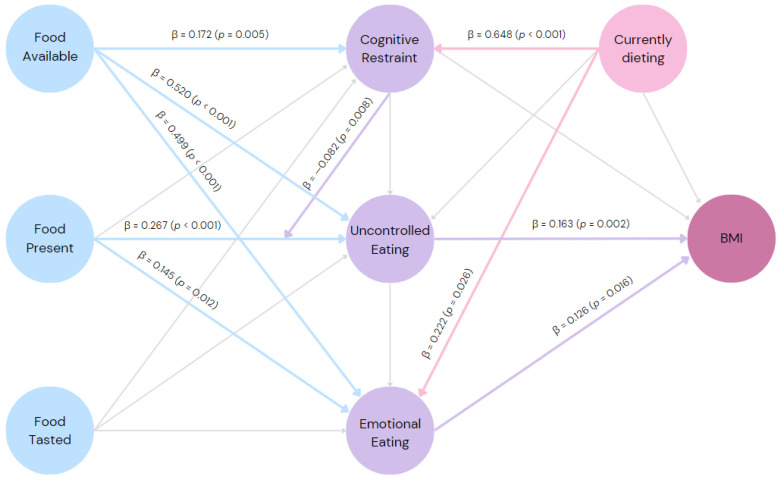
Final inner model: the predictive relationship between the psychological impact of the food environment, aspects of eating behavior, current dieting, and body mass index of 413 adults living in urban areas; β = path coefficient values; *p*-values of the statistics were based on bootstraps with 5000 samples; blue arrows = paths emerging from PFS factors; violet arrows = paths emerging from TFEQ-R18 factors; light pink arrows = paths emerging from current dieting; light grey arrows = non-significant paths.

**Table 1 foods-13-00052-t001:** Sociodemographic characteristics of 413 Brazilian men and women participating in a cross-sectional data collection on eating behavior aspects and the psychological impact of food-rich environments.

Variables	*n* (%)	Variables	*n* (%)
Age (years old)		Sex	
18–30	312 (75.5%)	Women	205 (49.6%)
31–40	63 (15.3%)	Men	208 (50.4%)
41–50	21 (5.1%)	Education level	
51–60	17 (4.1%)	Incomplete primary education	1 (0.2%)
Gender		Completed primary education	2 (0.5%)
Female	204 (49.4%)	Incomplete high school	3 (0.7%)
Male	208 (50.4%)	Completed high school	34 (8.2%)
Other: nonbinary	1 (0.2%)	Incomplete higher education	145 (35.1%)
Race		Completed higher education	140 (34%)
White	333 (80.6%)	Postgraduate	88 (21.3%)
Black	16 (3.9%)	Monthly family income	
Mixed	49 (11.9%)	Up to BRL 1200.00	9 (2.2%)
Asian	15 (3.6%)	BRL 1200.01 to BRL 3600.00	90 (21.8%)
		BRL 3600.01 to BRL 7200.00	132 (32%)
Region		BRL 7200.01 to BRL 10,800.00	63 (15.2%)
Northeast	6 (1.45%)	BRL 10,800.01 to BRL 14,000.00	48 (11.6%)
Central west	6 (1.45%)	Higher than BRL 14,000.01	71 (17.2%)
Southeast	394 (95.4%)	Dieting frequency	
South	7 (1.7%)	Never	112 (27.1%)
		Rarely	125 (30.3%)
BMI classification		Sometimes	94 (22.7%)
Underweight (<18.5 kg/m^2^)	14 (3.4%)	Often	61 (14.8%)
Normal weight (18.5–24.9 kg/m^2^)	240 (58.1%)	Always	21 (5.1%)
Overweight (25.0–29.9 kg/m^2^)	111 (26.9%)	Currently dieting	
Obesity (≥30.0 kg/m^2^)	47 (11.4%)	Yes	95 (23%)
Missing	1 (0.2%)	No	318 (77%)

BMI = body mass index; USD 1.00 = BRL 5.32 in November 2022.

**Table 2 foods-13-00052-t002:** Mean values, standard deviation, and factor loadings of the Power of Food Scale (PFS) indicators of a sample of 413 Brazilian adults living in urban areas.

PFS Indicators	Mean ± SD	Factor Loadings
Food Available (composite reliability = 0.853; AVE = 0.574)	-	-
PFS 1	2.5 ± 1.2	0.715
PFS 2	2.2 ± 1.1	0.678
PFS 5	2.2 ± 1.3	0.751
PFS 10	2.5 ± 1.2	0.692
PFS 11	2.2 ± 1.3	0.814
PFS 13	1.9 ± 1.1	0.849
Food Present (composite reliability = 0.785; AVE = 0.599)	-	-
PFS 3	3.6 ± 1.1	0.675
PFS 4	3.1 ± 1.3	0.696
PFS 6	2.9 ± 1.3	0.805
PFS 7	2.6 ± 1.3	0.733
Food Tasted (composite reliability = 0.718; AVE = 0.453)	-	-
PFS 8	2.7 ± 1.3	0.676
PFS 9	3.6 ± 1.2	0.606
PFS 12	2.5 ± 1.2	0.694
PFS 14	3.4 ± 1.2	0.576
PFS 15	3.0 ± 1.3	0.498

AVE = average variance extracted; SD = standard deviation; #PFS is copyrighted by Drexel University, copies of the PFS can be obtained by writing to Prof. Michael Lowe: lowe@drexel.edu.

**Table 3 foods-13-00052-t003:** Mean values, standard deviation, and factor loadings of the Three-Factor Eating Questionnaire (TFEQ) indicators of a sample of 413 Brazilian adults living in urban areas.

TFEQ-R18 Indicators	Mean ± SD	Factor Loadings
Cognitive Restraint (composite reliability = 0.847; AVE = 0.516)	-	-
TFEQ 2	1.9 ± 0.9	0.835
TFEQ 11	2.2 ± 0.9	0.861
TFEQ 12	1.9 ± 0.9	0.767
TFEQ 15	2.4 ± 0.9	0.564
TFEQ 16	2.5 ± 0.8	0.444
TFEQ 18	3.4 ± 1.7	0.654
Uncontrolled Eating (composite reliability = 0.842; AVE = 0.435)	-	-
TFEQ 1	2.3 ± 0.9	0.529
TFEQ 4	2.1 ± 0.9	0.825
TFEQ 5	2.3 ± 0.8	0.635
TFEQ 7	2.2 ± 0.8	0.578
TFEQ 8	2.1 ± 0.9	0.695
TFEQ 9	2.0 ± 0.9	0.720
TFEQ 13	1.9 ± 0.8	0.652
TFEQ 14	2.0 ± 0.7	0.517
TFEQ 17	2.2 ± 0.8	0.731
Emotional Eating (composite reliability = 0.807; AVE = 0.836)	-	-
TFEQ 3	2.6 ± 1.0	0.849
TFEQ 6	2.2 ± 1.0	0.887
TFEQ 10	1.8 ± 0.9	0.804

AVE = average variance extracted; SD = standard deviation.

**Table 4 foods-13-00052-t004:** Pearson’s correlations (and *p*-values) of PFS and TFEQ factors in a Brazilian adult sample.

Variables	Food Available	Food Present	Food Tasted	Cognitive Restraint	Uncontrolled Eating
Food Available	1.00	-	-	-	-
Food Present	0.61 (<0.001)	1.00	-	-	-
Food Tasted	0.54 (<0.001)	0.59 (<0.001)	1.00	-	-
Cognitive Restraint	0.12 (0.01)	0.03 (0.48)	0.02 (0.63)	1.00	-
Uncontrolled Eating	0.68 (0.001)	0.59 (<0.001)	0.45 (<0.001)	0.02 (0.63)	1.00
Emotional Eating	0.60 (0.001)	0.46 (<0.001)	0.38 (<0.001)	0.19 (<0.001)	0.52 (<0.001)

**Table 5 foods-13-00052-t005:** Mean values and standard deviation of aspects of eating behavior and psychological impact of food-rich environments according to proximity levels of palatable foods in Brazilian women and men living in urban areas.

Variables	Women (*n* = 205) Mean ± SD	Men (*n* = 208) Mean ± SD	*p*-Value	Cohen’s d
Cognitive Restraint	2.5 ± 0.7	2.3 ± 0.7	<0.001	0.342
Uncontrolled Eating	2.1 ± 0.6	2.1 ± 0.6	0.920	0.010
Emotional Eating	2.5 ± 0.9	2.0 ± 0.8	<0.001	0.532
Food Available	2.4 ± 0.9	2.1 ± 0.8	<0.001	0.391
Food Present	3.3 ± 0.9	2.8 ± 0.9	<0.001	0.450
Food Tasted	3.1 ± 0.8	2.9 ± 0.8	0.024	0.222
PFS Aggregated Factor	2.9 ± 0.8	2.6 ± 0.7	<0.001	0.420

SD = standard deviation.

**Table 6 foods-13-00052-t006:** Differences between men’s and women’s path coefficients for predicting the effect of the psychological impact of food-rich environments on aspects of eating behavior in a Brazilian urban sample.

Paths	Mean Difference (Men–Women) Path Coefficient	*p*-Value
Cognitive Restraint → BMI	0.013	0.904
Cognitive Restraint → Uncontrolled Eating	−0.053	0.461
Currently Dieting → Cognitive Restraint	0.307	0.192
Currently Dieting → Emotional Eating	−0.047	0.819
Currently Dieting → BMI	0.132	0.538
Currently Dieting → Uncontrolled Eating	−0.153	0.360
Emotional Eating → BMI	0.040	0.697
Food Tasted → Cognitive Restraint	0.088	0.546
Food Tasted → Emotional Eating	−0.147	0.137
Food Tasted → Uncontrolled Eating	0.041	0.648
Food Available → Cognitive Restraint	0.014	0.914
Food Available → Emotional Eating	−0.006	0.957
Food Available → Uncontrolled Eating	0.075	0.389
Food Present → Cognitive Restraint	−0.150	0.272
Food Present → Emotional Eating	0.090	0.457
Food Present → Uncontrolled Eating	−0.120	0.202
Uncontrolled Eating → BMI	−0.021	0.831
Cognitive Restraint × Food Present → Uncontrolled Eating	−0.084	0.160

BMI = body mass index.

## Data Availability

Data is contained within the article.

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
