# Peer review of "The Psychological Impact of the Widespread Availability of Palatable Foods Predicts Uncontrolled and Emotional Eating in Adults"

_foods, 2023, doi:10.3390/foods13010052_

Round 1

Reviewer 1 Report

Comments and Suggestions for Authors

The study addresses an important and timely topic regarding the psychological impact of environments rich in palatable foods on eating behavior, specifically cognitive restraint (CR), uncontrolled eating (UE), and emotional eating (EE). The use of the Power of Food Scale (PFS) and Three-Factor Eating Questionnaire (TFEQ-18) for assessment, along with the application of partial least squares structural equation modeling (PLS-SEM), adds rigor to the methodology. However, several aspects need to be considered for improvement and clarification:

The abstract outlines the study's aim but could be more explicit in presenting the specific hypotheses tested. Clearly stating the hypotheses and objectives will help readers understand the focus of the study from the outset.

Introduction: In this study, the introduction provides a good overview of the background; it could benefit from a more concise articulation of the research gap and a specific statement of the research questions or hypotheses. The relationship between non-homeostatic factors and eating behavior is mentioned, but a clearer connection to the study's focus on palatable food environments and psychological impact would enhance the introduction.

Theoretical Background and Model: The manuscript could benefit from a more explicit theoretical framework that guides the study. What theoretical perspectives or models underpin the exploration of the psychological impact of food-rich environments on eating behavior? This would add depth to the rationale for the study.

Methodology: Clarify the sampling strategy and characteristics of the online sample. Discuss the representativeness of the sample and acknowledge any potential biases associated with the online data collection method. Provide more information on the demographics of the sample, including age, gender distribution, and any other relevant characteristics that may influence eating behavior. Discuss the instrument of the 8-point Likert scale.

Results: Consider providing effect sizes for key findings to help readers assess the practical significance of the relationships observed.

Discussion: The discussion is comprehensive, but it could be more concise. Consider streamlining the presentation of results and focusing on the key findings that directly address the research questions. Address potential limitations more explicitly, including the cross-sectional design, and discuss how these limitations may impact the generalizability of the findings. Suggest directions for future research based on the study's limitations and potential areas for further exploration.

Conclusion: The conclusion should succinctly recapitulate the main findings and their implications for the field. Consider restating the practical implications for health strategies and interventions.

Include a section discussing ethical considerations, especially given the online data collection method. Address issues such as informed consent, privacy, and any steps taken to ensure the ethical conduct of the study.

It would be beneficial to explore avenues for future research based on the current findings.

Clarify how the study's results may inform practical interventions or policies aimed at improving eating behavior in light of the identified associations.

The writing is generally clear, but some sentences are complex, and simplifying them may enhance readability.

Ensure consistency in terminology and use of acronyms throughout the abstract.

The study contributes valuable insights into the psychological impact of food environments on eating behaviors, emphasizing the importance of considering environmental factors in health interventions. However, addressing these points will strengthen the manuscript and contribute to its overall impact in the field. Therefore, the study requires some minor revisions for further consideration

Comments on the Quality of English Language

The writing is generally clear, but some sentences are complex, and simplifying them may enhance readability.

Author Response

Reviewer #1

1) The study addresses an important and timely topic regarding the psychological impact of environments rich in palatable foods on eating behavior, specifically cognitive restraint (CR), uncontrolled eating (UE), and emotional eating (EE). The use of the Power of Food Scale (PFS) and Three-Factor Eating Questionnaire (TFEQ-18) for assessment, along with the application of partial least squares structural equation modeling (PLS-SEM), adds rigor to the methodology. However, several aspects need to be considered for improvement and clarification

R: We would like to thank the reviewer for the careful reading and suggestions.

2) The abstract outlines the study's aim but could be more explicit in presenting the specific hypotheses tested. Clearly stating the hypotheses and objectives will help readers understand the focus of the study from the outset.

R: We agree with the reviewer. We have included the hypotheses in the abstract, as follows:

“The hypotheses were: a) the psychological impact (i.e., motivation to eat) of an environment rich in palatable foods will positively predict CR, UE and EE; b) Dieting will predict CR, UE and EE; c) CR, UE and EE will positively predict body mass index (BMI).”

3) Introduction: In this study, the introduction provides a good overview of the background; it could benefit from a more concise articulation of the research gap and a specific statement of the research questions or hypotheses. The relationship between non-homeostatic factors and eating behavior is mentioned, but a clearer connection to the study's focus on palatable food environments and psychological impact would enhance the introduction.

R: We agree with the reviewer. The introduction was changed.

4) Theoretical Background and Model: The manuscript could benefit from a more explicit theoretical framework that guides the study. What theoretical perspectives or models underpin the exploration of the psychological impact of food-rich environments on eating behavior? This would add depth to the rationale for the study

R: We agree with the reviewer. We included some references that grounded our study. We included a sentence in the start of this section.

5) Methodology: Clarify the sampling strategy and characteristics of the online sample. Discuss the representativeness of the sample and acknowledge any potential biases associated with the online data collection method. Provide more information on the demographics of the sample, including age, gender distribution, and any other relevant characteristics that may influence eating behavior. 

R: We agree with the reviewer. 

The bias about online data collection was described in the limitations section.

The demographics was included in table 1 in the results section.

8) Discuss the instrument of the 8-point Likert scale.

R: The TFEQ-R18 has a question (the last question) measured by a 8-point scale (please see: https://www.med.umich.edu/pdf/weight-management/TFEQ-r18.pdf). We included a sentence to clarify it, as follows:

“For the last item of the TFEQ-R18, the participant used an 8-point scale for restraint (1- no restraint in eating to 8- total restraint). “

9)Results: Consider providing effect sizes for key findings to help readers assess the practical significance of the relationships observed.

R: We agree with the reviewer about the importance of effect sizes. However, we do not understand the suggestion as all effect sizes have already been reported.

For PLS-SEM, the effect sizes were measured by f² (Section 4.3). Pearson’s r (Table 4) was used for the correlation and Cohen’s d (Table 5) for the group comparisons.

10) Discussion: The discussion is comprehensive, but it could be more concise. Consider streamlining the presentation of results and focusing on the key findings that directly address the research questions. Address potential limitations more explicitly, including the cross-sectional design, and discuss how these limitations may impact the generalizability of the findings. Suggest directions for future research based on the study's limitations and potential areas for further exploration.

R: We agree with the reviewer. We included a limitation section in the discussion. 

11) Conclusion: The conclusion should succinctly recapitulate the main findings and their implications for the field. Consider restating the practical implications for health strategies and interventions.

R: We agree with the reviewer. The conclusion has been rewritten to state the aims of the study.

12) Include a section discussing ethical considerations, especially given the online data collection method. Address issues such as informed consent, privacy, and any steps taken to ensure the ethical conduct of the study.

R: The ethical section and statements are at the end of section 3.1

13) It would be beneficial to explore avenues for future research based on the current findings.

R: Future research was discussed at the end of limitations section. 

14) The writing is generally clear, but some sentences are complex, and simplifying them may enhance readability. Ensure consistency in terminology and use of acronyms throughout the abstract.

R: We proofread the text and abstract.

15) The study contributes valuable insights into the psychological impact of food environments on eating behaviors, emphasizing the importance of considering environmental factors in health interventions. However, addressing these points will strengthen the manuscript and contribute to its overall impact in the field. Therefore, the study requires some minor revisions for further consideration.

R: We appreciate all your suggestions.

Reviewer 2 Report

Comments and Suggestions for Authors

Dear Authors,

The manuscript (foods-2782612) submitted for review is interesting.

Authors, Please note and address the following comments:

Introduction

The introduction should be better organized and end with the aim of the work. For now, it's chaos, although the information presented in this part of the manuscript is interesting. The topic of the manuscript is interesting.

Material and Methods/ Results/ Discussion/ Conclusion

The sections such as material and methods, results, discussion, as well as conclusion are  well written.

Authors wrote that participants were recruited via social media (e.g., Instagram and Facebook) and in 186 person on the university campus. I have a question, How was the recruitment of respondents carried out via Instagram and Facebook? Was it a university or private account? Who recruited?

The group of respondents seems to be quite small to draw this type of conclusions, especially since the group of respondents was accidental and not random.

If there are any limitations to the results of this study, it is a good idea to add them to the manuscript. Limitation of this study should be added as a separate chapter.

References:

References are not cited according to journal rules. Publications from MDPI provide information on how to properly cite. Authors may also find this information in the authors' guide.

 Now it is:

9. Eating when depressed, anxious, bored, or happy: Are emotional eating types associated with unique psychological and phys- 529 ical health correlates? Appetite. 2018;125:410-417. doi:10.1016/J.APPET.2018.02.022

It should be like this:

Braden A, Musher-Eizenman D, Watford T, Emley E. Eating when depressed, anxious, bored, or happy: Are emotional eating types associated with unique psychological and physical health correlates? Appetite. 2018 Jun 1;125:410-417. doi: 10.1016/j.appet.2018.02.022. Epub 2018 Feb 22. PMID: 29476800.

I believe that this manuscript concerns an important area of research in an international context. However, it should be noted that these are preliminary studies. I recommend that the manuscript needs revision.

Reviewer

Author Response

Reviewer #2

Dear Authors,

The manuscript (foods-2782612) submitted for review is interesting.

Authors, Please note and address the following comments:

1) Introduction: The introduction should be better organized and end with the aim of the work. For now, it's chaos, although the information presented in this part of the manuscript is interesting. The topic of the manuscript is interesting.

R: We agree with the reviewer. We have made many changes in the introduction section. 

2) Material and Methods/ Results/ Discussion/ Conclusion: The sections such as material and methods, results, discussion, as well as conclusion are  well written.

R: We would like to thank you for the revision and suggestions

3) Authors wrote that participants were recruited via social media (e.g., Instagram and Facebook) and in person on the university campus. I have a question, How was the recruitment of respondents carried out via Instagram and Facebook? Was it a university or private account? Who recruited?

R: We included more information about the recruitment process, as follows:

“Participants were recruited online via social media (e.g., Instagram and Facebook) and in person on the university campus. An advertising banner with QR code and direct message was used for recruitment. The entire recruitment was carried out by the authors of the study. “

4) The group of respondents seems to be quite small to draw this type of conclusions, especially since the group of respondents was accidental and not random.

Response: In this case it is not possible to draw a random sample, because in a random sample everyone has the same chance of being selected for the study. A random sample is only used to test a specific population or scenario. Also, a small sample would result in a type 2 (beta) error and not alpha. The sample therefore has sufficient power to detect differences between men and women and to identify associations.

5) If there are any limitations to the results of this study, it is a good idea to add them to the manuscript. Limitation of this study should be added as a separate chapter.

R: We moved the limitation paragraph for a limitation section, as suggested.

6) References: References are not cited according to journal rules. Publications from MDPI provide information on how to properly cite. Authors may also find this information in the authors' guide.

 Now it is:

  1. Eating when depressed, anxious, bored, or happy: Are emotional eating types associated with unique psychological and phys- 529 ical health correlates? Appetite. 2018;125:410-417. doi:10.1016/J.APPET.2018.02.022

It should be like this:

Braden A, Musher-Eizenman D, Watford T, Emley E. Eating when depressed, anxious, bored, or happy: Are emotional eating types associated with unique psychological and physical health correlates? Appetite. 2018 Jun 1;125:410-417. doi: 10.1016/j.appet.2018.02.022. Epub 2018 Feb 22. PMID: 29476800.

R: We agree with the reviewer. We have used Mendeley to avoid problems. However, as some errors remained, we have corrected all references manually.

7) I believe that this manuscript concerns an important area of research in an international context. However, it should be noted that these are preliminary studies. I recommend that the manuscript needs revision.

R: We appreciate your suggestions